# Mie-Metamaterials-Based Electromagnetic Absorbing Concrete

**Hongya Wu [1,2,3], Han Dong [1], Yafan Zhang [1], Danni Zhou [1], Hao Fang [1], Shengjian Qin [1], Guoqiang Qin [1] and Guanglei Zhang [1,2,*]**

1 School of Materials Science and Engineering, Shijiazhuang Tiedao University, Shijiazhuang 050043, China
2 Engineering Research Center of Metamaterials & Micro-Devices, Shijiazhuang 050043, China
3 Shaanxi Key Laboratory of Artificially-Structured Functional Materials and Devices, Xi'an 710049, China
* Correspondence: zhgl@stdu.edu.cn

**Abstract:** A metamaterial structure using steel reinforcement and dielectric cubes was designed, which was introduced into concrete to improve the electromagnetic wave absorption performance of concrete. A near-unit absorption peak at 9.82 GHz was generated in simulation and experimental results. It is also demonstrated that this metamaterial absorbing concrete is insensitive to polarization and can operate at wide angle of incidence.

**Keywords:** metamaterials; absorption; absorbing concrete

## 1. Introduction

With the rapid development of electronic information technology, people's attention has been attracted by electromagnetic interference. The extensive use of electromagnetic waves (EMW) is potentially harmful to human health, electronic information, and even the national information security [1,2]. Recently, the negative effect of electromagnetic radiation has been the focus of social attention; the development of EMW absorbing materials, which can be used in microwave interference protection, microwave darkroom, and other military and civilian fields, is of great significance [3]. As the most commonly used construction material globally, the absorbing performance of cement-based materials is limited. To improve the EMW absorption capacity of concrete, various absorbents have been studied by researchers and scientists, such as ore tailings [4], fly ash [5], and ferrite and steel fiber [6], which can be used to enhance the electromagnetic absorption capacity of the cementitious material. At the same time, nanomaterials [7], chiral materials, conductive polymers, and ceramics were also explored as absorbents [8,9]. The mechanical properties and durability of cement-based material will be negatively affected to a certain extent by the EMW absorbing additives, which limit the development of cement-based, wave-absorbing materials [10].

Enormous progress in the field of metamaterials EMW absorption has been made in the past few years. The extraordinary electromagnetic properties of metamaterials [11,12] leads to its potential application for metamaterial absorbers, which are thin and small. They also have a simple structure and show high absorption capacity, which can be used for electromagnetic stealth [13,14]. Mie-resonance-based metamaterial has been deeply studied in recent years. In Mie-resonance-based metamaterial, the electromagnetic interaction of dielectric particles with relative high permittivity can exhibit a strong magnetic or electric resonance based on Mie theory. From the oscillation of resulting magnetic or electric dipole, negative permeability or permittivity can be produced [15,16], and the frequency of magnetic or electric resonance shifts with shapes, sizes, dielectric constant, and permeability of the dielectric cubes [17]. Mie-resonance-based metamaterials with single-band [18], dual-band [19], multi-band [20] and broad-band [21] EMW absorption are widely studied in microwaves, and almost 100% absorption bands of Mie-resonance-based metamaterial absorbers with a wide incident angle and polarization insensitivity was reported [18].

Here, an alternative approach of absorbing concrete in the idea of combining cement-based materials in civil engineering with Mie-resonant-metamaterial absorbers is studied,

where steel bars in the cement-based material are combined with CaTiO$_3$ ceramic cubes to obtain near-unity absorption. The absorption frequency of the cementitious material can be easily tuned by the structure of the ceramic particles and steel bars.

## 2. Materials and Methods

### 2.1. Materials

The concrete was made with standard Portland cement P.O 42.5. The silica sand (SS) was introduced as the fine filler aggregate, and its particle size ranges from 0.2 to 0.4 mm. The aggregate, with a diameter of 5–20 mm, was used in the preparation of concrete. Plain round bars (steel-1010, the conductivity of which is $\sigma = 6.993 \times 106$ S/m.) with diameter of 8 mm were used to prepare reinforced concrete as well as part of the metamaterial absorbers. CaTiO$_3$ ceramics were prepared with analytical pure calcium carbonate and titanium dioxide, which are commercial products.

### 2.2. Processing

The CST Microwave Studio was used to simulate the microwave properties. The sizes and material parameters for simulations are in agreement with the experimental values. The metamaterials absorber was designed by simulation through adjusting the structure, size, and materials parameters.

A calcium titanate sample was synthesized by the solid-state reaction method. The mixer of stoichiometric quantities of calcium carbonate and titanium dioxide were put into a polymeric amide bottle to ball-mill for 10 h with ethanol and zirconium oxide balls. Then, an oven was used to dry the obtained slurry at 120 °C, followed by calcining at 1150 °C for 120 min in an alumina crucible. After ball-milling a second time, the obtained powders were pressed into dimensions of 30 mm × 20 mm × 3 mm under a pressure of 5 MPa. A muffle furnace (KSL-1700X, KMT, China) was used to sinter the compacted powders at 1350 °C for 2 h. The permittivity of the obtained ceramics was 150 + 0.002 i. To prepare dielectric particles, the ceramics were cut into 2.50 mm cubes which were stick to the center of the crossbar.

The proportions of the concrete are demonstrated in Table 1. The fresh mortar was cast into 150 mm × 150 mm × 150 mm plate molds, flattened, and compacted on a vibrating table for 10 s. Then the metamaterials absorber was put into the center of the concrete; after standing for 24 h, the molds were removed, and the samples were put into a curing room for 28 days of standard curing. Finally, the obtained concrete was cut into 125 mm × 125 mm × 125 mm to manufacture the designed EMW absorbing concrete sample, as shown in Figure 1.

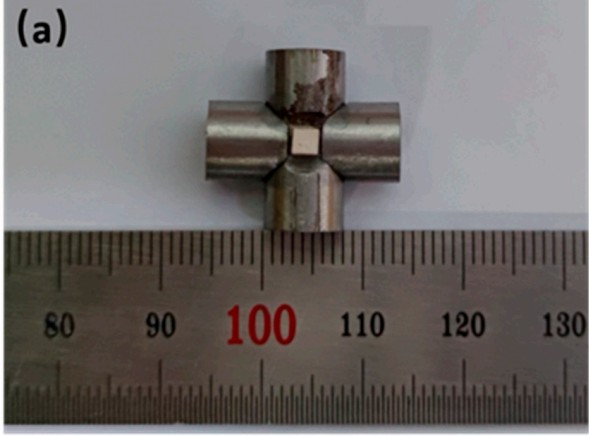
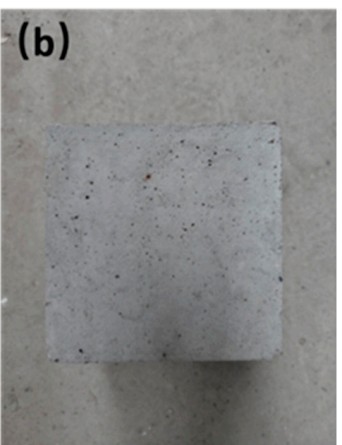

**Figure 1.** Wave-absorbing concrete samples: (**a**) unit cell of the metamaterial absorber sample; (**b**) optical image of EMW absorbing concrete sample.

**Table 1.** Mix proportions of the concrete.

| Cement (kg/m$^3$) | Sand (kg/m$^3$) | Stone (kg/m$^3$) | Water (kg/m$^3$) |
|---|---|---|---|
| 408 | 576 | 1169 | 200 |

### 2.3. Characterization

The microwave properties (S11, S21) of the designed samples were measured by the E5063A vector network analyzer and standard gain horn antenna XB-HA90-20N, ranging from 8 GHz to 12 GHz. The absorptivity can be calculated from the test results with Equation (1).

$$A(f) = 1 - |S_{11}(f)|^2 - |S_{21}(f)|^2 \tag{1}$$

## 3. Results

The unit cell and dimensions of the structure are shown in Figure 2a,b where the dimensions of the steel bar are 18 mm in length and 8 mm in diameter. The sizes and the material parameters for the simulations are the same as the experimental ones. The negligible free-space path loss was not considered in simulations.

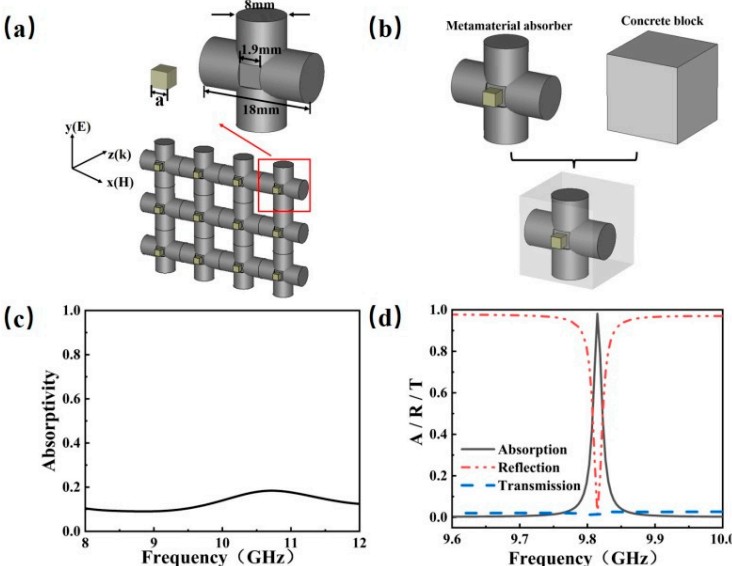

**Figure 2.** The designed metamaterial absorber and simulation results: (**a**) unit cell of the steel bar structure, in the simulations, the EMW k is incident to the *y*-axis, with magnetic field H parallel to the *z*-axis, and electric field E parallel to the *x*-axis; (**b**) unit cell of the metamaterial absorber within the concrete; (**c**) simulated absorptivity spectrum of concrete without metamaterial absorber; and (**d**) simulated absorptivity spectrum when a = 2.50 mm, $\varepsilon$ = 150.

The simulated result is shown in Figure 2c,d; it can be seen from Figure 2c that the absorptivity of concrete without metamaterial absorber is about 0.1, which is low. When the metamaterial absorber is introduced into concrete as in Figure 2b, the absorptivity at 9.82 GHz reaches 98%, as shown in Figure 2d. These results indicate that the EMW absorbance is caused by metamaterial structure.

In the experiment, we measured S-parameter curves of metal plane reflector for S11 and free-space path without samples for S21 which are named S11cal and S21cal, respectively. The measured S11 and S21 with the metamaterial absorber as sample were named S11sam and S21sam, respectively. Considering the loss of the environment (e.g., air), the two S-parameters need to be subtracted, and the final S-parameter is S = Ssam − Scal. Then, absorptivity can be calculated by Equation (1). The simulated and experimental results are shown in Figure 3. There are small differences between the measured and

simulated results which are caused by tolerances in the manufacture and environmental disturbance in experiment, et al. Figure 3a demonstrates the absorption both in simulated and experimental results. The effect of polarization angles φ and incident angles θ for transverse electric (TE) and transverse magnetic (TM) waves on the absorbance of the absorbers at 9.82 GHz are studied and the results are shown in Figure 3. The results indicate that the EMW absorption is insensitive to polarization because the absorption rate does not change with φ varying in the range of 0–75°, as shown in Figure 3b. For the TE mode, the absorption at 9.82 GHz decreases gradually with the increasing of θ from 0° to 75°, and reaches 90% when θ = 75°, and the results are shown in Figure 3c. For the TM mode, the absorption at 9.82 GHz decreases from 98 to 50% with the increasing of θ from 0° to 75°, which is shown in Figure 3d. When the TE wave is obliquely incident to the designed metamaterials absorber, the electric field direction is always parallel to the surface of the absorber; however, the angle between the magnetic field and the surface of the absorber increases with the increasing of incident angle, resulting in a decrease in absorption. The TM wave shows similar results with the TE wave. The simulation and experimental results demonstrate the good absorption performance of the designed absorbing concrete with wide incident angle.

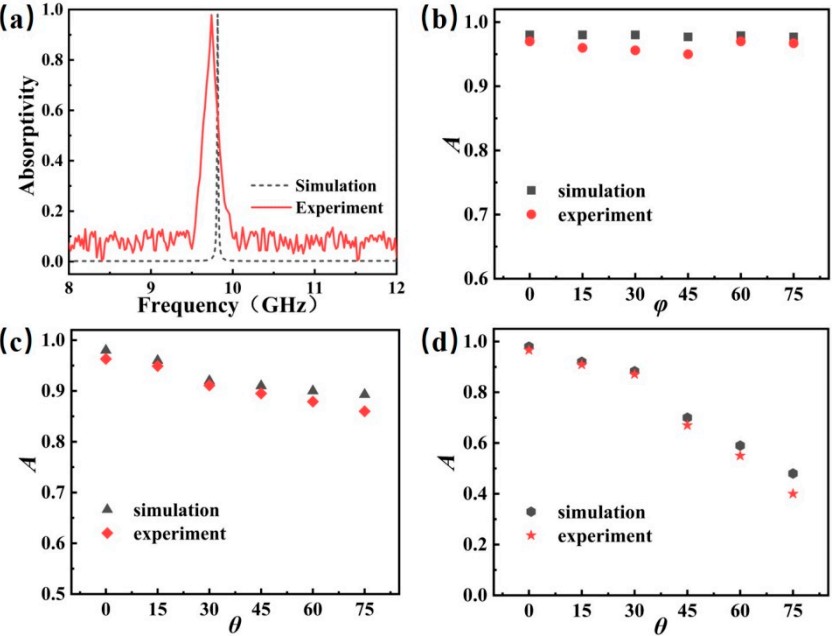

**Figure 3.** Simulation and experimental results of EMW absorption performance of the designed absorbing concrete: (**a**) the simulation and experimental absorption spectra; (**b**) absorption with different φ at 9.82 GHz; (**c**) absorptivity with different θ for TE mode at 9.82 GHz; and (**d**) absorptivity with different θ of for TM mode at 9.82 GHz.

To clarify the origin of the EMW absorption, the energy density of electric field and magnetic field at 9.82 GHz when a = 2.5 mm was studied, and the results are shown in Figure 4. Figure 4a,c show the cross-sectional E-field and H-field distribution in the y–z plane and x–z plane, respectively. Figure 4a,c demonstrate a resonance inside the dielectric cubes which comes from the interaction of the dielectric cube with the incident electromagnetic waves. At 9.82 GHz, a toroidal electric field in the y–z plane and a horizontal magnetic field in the *x*-direction were induced inside the dielectric cubes, and the electromagnetic field distribution indicates that the dielectric cubes are equivalent to a magnetic dipole, showing the characteristics of magnetic resonance. Figure 4b,d show the E-field and H-field energy density distributions of the dielectric cubes and the crossed steel bars, respectively. At 9.82 GHz, the E-field energy and H-field energy of the incident

electromagnetic wave are concentrated inside the dielectric cubes. Therefore, the EMW absorption was caused by the loss of the dielectric particles, which was a magnetic loss.

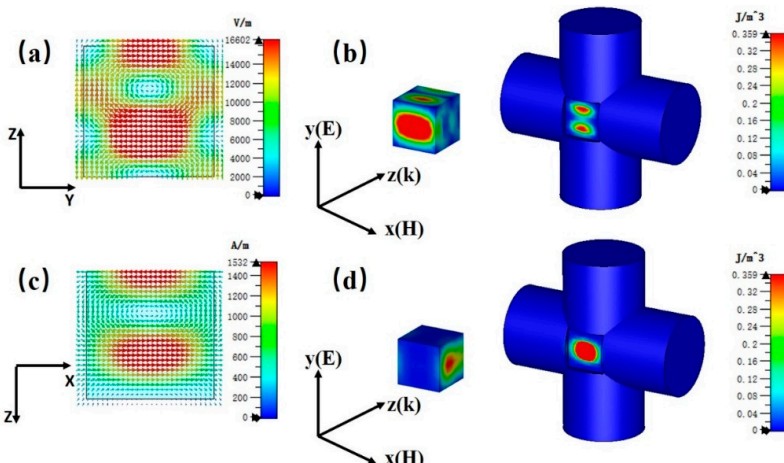

**Figure 4.** Simulated results of the distribution and energy density of electric field and magnetic field at 9.82 GHz: (**a**) E-field distribution; (**b**) E-field energy density; (**c**) M-field distribution; and (**d**) M-field energy density.

## 4. Discussion

As the electromagnetic properties of Mie-resonance based-metamaterials can be tailored by the shape and size of the dielectric cubes, the EMW absorbing concrete can be designed to work at different frequencies. The simulated S11 and S21 in dB of the designed structure with "a" ranging from 2.4 mm to 2.6 mm and the dielectric constant $\varepsilon$ of the dielectric cubes ranging from 146 to 150 is shown in Figure 5. Further, the calculated absorptivity is shown in Figure 6.

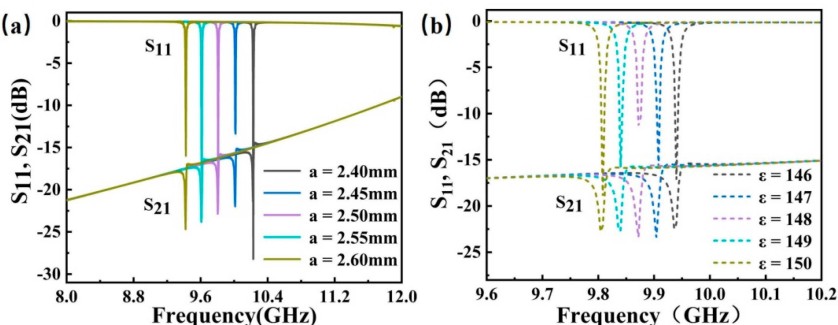

**Figure 5.** (**a**) simulated S11 and S21 spectra when a = 2.40–2.60 mm, $\varepsilon$ = 150; and (**b**) simulated S11 and S21 spectra when a = 2.50 mm, $\varepsilon$ = 146–150.

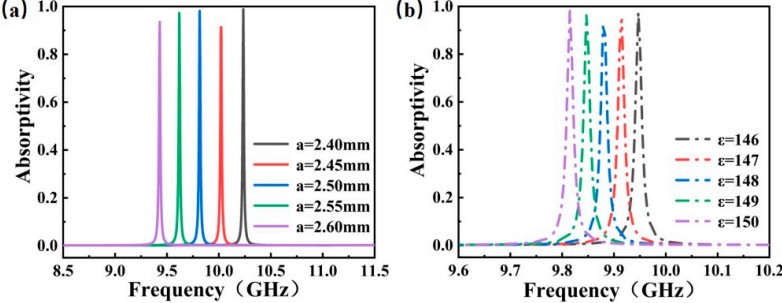

**Figure 6.** (**a**) simulated absorptivity spectra when a = 2.40–2.60 mm, $\varepsilon$ = 150; and (**b**) simulated absorptivity spectra when a = 2.50 mm, $\varepsilon$ = 146–150.

The results indicate that the working frequency shifts with the dimensions of the structure and the permittivity of the dielectric cubes. The reflectivity reaches −15 dB, the transmission is low in the study frequency range, and the absorptivity is higher than 90%. According to these results, metamaterials based EMW absorbing concrete can be designed to work at different frequencies to work in different environments.

In addition, the recent literature on metamaterial absorbers and EMW absorbing concrete was surveyed. Further, four representative papers were selected to compare with our work, which are shown in Table 2. The results in Table 2 indicate that the proposed metamaterial-based EMW absorbing concrete has high absorptivity and a simple structure.

**Table 2.** A comparison between the proposed work and existing works.

| Absorber | Working Frequency | Absorptivity | S11 |
|---|---|---|---|
| dielectric-bar | 9.82 GHz | 98% | |
| metal–dielectric–ground [22] | 14.15 GHz | 84% | |
| metal–dielectric [23] | 95 GHz | 40% | |
| Nano-$Fe_3O_4$ magnetic fluid [24] | 17 GHz | | −35 dB |
| expanded perlite [25] | 4.5–18 GHz | | −10 dB |

## 5. Conclusions

A wave-absorbing concrete based on Mie resonance was simulated and experimentally constructed. An absorption peak at 9.82 GHz with a near-unit absorption rate was obtained. At the absorption frequency of 9.82 GHz, a Mie resonance was induced inside the dielectric cubes, resulting in simulated and experimental absorption rates of 98% and 97%, respectively. The main loss of the EMW at the resonant frequencies came from the magnetic loss based on the Mie resonance. The experimental and simulated results are in general agreement. The method of combining Mie-resonance-based metamaterials with concrete provides a new idea for wave absorbing concrete.

**Author Contributions:** Conceptualization, H.W. and G.Z.; methodology, H.W.; validation, H.W., H.D. and Y.Z.; formal analysis, H.W. and H.D.; investigation, H.D. and Y.Z.; resources, H.W., H.D. and S.Q.; data curation, H.D., Y.Z., D.Z., H.F. and G.Q.; writing—original draft preparation, H.W. and H.D.; writing—review and editing, H.W., H.D., Y.Z., D.Z., H.F. and G.Q.; visualization, S.Q.; supervision, H.W. and G.Z.; project administration, H.W. and G.Z.; funding acquisition, H.W. and G.Z. All authors have read and agreed to the published version of the manuscript.

**Funding:** This research was funded by National Natural Science Foundation of China, grant number 51502179; Natural Science Foundation of Hebei Province, grant number E2020210076; Natural Science Foundation of Hebei Education Department, grant number BJ2021037; Fundamental Research Funds of Shaanxi Key Laboratory of Artificially-Structured Functional Materials and Devices, grant number AFMD-KFJJ-21214; Technological Innovation Guidance Plan Project of Hebei Province, grant number 20471101d.

**Institutional Review Board Statement:** Not applicable.

**Data Availability Statement:** Not applicable.

**Conflicts of Interest:** The authors declare no conflict of interest. The funders had no role in the design of the study; in the collection, analyses, or interpretation of data; in the writing of the manuscript; or in the decision to publish the results.

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
