# Peer review of "Mie-Metamaterials-Based Electromagnetic Absorbing Concrete"

_applsci, doi:10.3390/app122211389_

Round 1
Reviewer 1 Report
This manuscript presents a metamaterial absorbing structure for use inside structures made of concrete. For this reviewer, some points need clarification before this manuscript is considered for acceptation on this publication.
It is not clear on the text if the free-space path loss on the measurement setup and simulation was taken into account on the values found for the absorption. How the free-space path loss influences the absorption value? What is the value of the S21 of the measurement setup if no concrete block is between the antennas? A discussion on this topic should be added to the manuscript to clarify the obtained results.
For this reviewer, all figures where the magnitude value of the absorption (as defined at eq. (1)), reflection and transmission are better analyzed when the values are in dB, so a reader can infer how a certain device with a known sensibility in dBm (or another unity expressed in dB) is affected by the proposed concrete structure. The shown absorption value, for example at line 116, of 98% in absorption of the transmitted power, should also be expressed in dB.
A comparison, even if with simulated values, between a traditional concrete block and the proposed structure on the absorption would greatly improve the manuscript and better present the benefit of the proposed metamaterial structure.
A comparison – maybe a table – between the proposed structure with other types of absorbents on concrete would improve the manuscript.
Line 107 – The authors used a ha340 antenna (waveguide type wr340), that operates between 2.2 and 3.3 GHz (D-Band). How can one use this antenna for an experiment operating around 10 GHz?
The manuscript presents some grammar errors, an English revision is advised.
Author Response
Dear Editor and Referee,
Thank you very much for the appropriate comments and valuable suggestions on our manuscript. According to referee's opinions, we make a major revision on the manuscript, and would like to provide following response:
The responds to the reviewer’s comments are as following:
- It is not clear on the text if the free-space path loss on the measurement setup and simulation was taken into account on the values found for the absorption. How the free-space path loss influences the absorption value? What is the value of the S21 of the measurement setup if no concrete block is between the antennas? A discussion on this topic should be added to the manuscript to clarify the obtained results.
Response: According to your suggestion, we added “The negligible free-space path loss was not considered in simulations.” in our manuscript. (Page 3/Line 117)
In the experiment, we measured S-parameter curves of metal plane reflector for S11 and free-space path without samples for S21 which are named S11cal and S21cal respectively. The measured S11 and S21 with the metamaterial absorber as sample were named S11sam and S21sam respectively. Considering the loss of the environment (e.g. air), the two S parameters need to be subtracted, and the final S parameter is S = Ssam − Scal. Then, absorptivity can be calculated by Eq.(1). We added these in our manuscript. (Page 4/Line 130)
- For this reviewer, all figures where the magnitude value of the absorption (as defined at eq. (1)), reflection and transmission are better analyzed when the values are in dB, so a reader can infer how a certain device with a known sensibility in dBm (or another unity expressed in dB) is affected by the proposed concrete structure. The shown absorption value, for example at line 116, of 98% in absorption of the transmitted power, should also be expressed in dB.
Response: According to your suggestion, we added S11 and S21 spectra in dB in our manuscript as Fig. 5. (Page 5/Line 180). But Linear data is required for calculations using eq. (1) and the results are magnitude values, so the absorption was not expressed in dB.
- A comparison, even if with simulated values, between a traditional concrete block and the proposed structure on the absorption would greatly improve the manuscript and better present the benefit of the proposed metamaterial structure.
Response: The simulated absorptivity spectrum of concrete without metamaterial absorber is added in our manuscript as Fig.2(c). It can be seen from Fig.2(c) that the absorptivity of concrete without metamaterial absorber is about 0.1 which is low. When the metamaterial absorber is introduced into concrete as Fig.2(b), the absorptivity at 9.82 GHz reaches 98 % as shown in Fig.2(d). These results indicate that the EMW absorbance is caused by metamaterial structure. (Page 4/Line 125)
- A comparison – maybe a table – between the proposed structure with other types of absorbents on concrete would improve the manuscript.
Response: According to your suggestion, we added a table to compare the proposed structure with other types of absorbents on concrete. ( Page 6/Line 195)
- Line 107 – The authors used a ha340 antenna (waveguide type wr340), that operates between 2.2 and 3.3 GHz (D-Band). How can one use this antenna for an experiment operating around 10 GHz?
Response: We made a mistake in writing and have changed the antenna to XB-HA90-20N. (Page 3/Line 109)
- The manuscript presents some grammar errors, an English revision is advised.
Response: We have revised the English and the revised parts in the manuscript are in red color.
We have tried our best to improve the manuscript and made some changes in it. These changes will not influence its content and framework. We look forward to your further response. Thanks again for your comments!
With best regards,
Yours sincerely,
Hongya Wu
Reviewer 2 Report
This manuscript highlights the design of a metamaterial structure using steel reinforcement and dielectric cubes introduced to the concrete. This paper states the improvement in the absorption of electromagnetic waves through these concrete samples. This manuscript is well-written and highlights the development of dielectric stealth materials designed to protect against microwave interference. However, based on my review I would suggest the authors revise this article to elaborate on the discussion (section 4).
Some key points to be considered are:
1. Define Mie metamaterials in the introduction section?
2. Experimental section, Explain the addition or placement of metamaterial absorber material within the concrete. A schematic may be added to display the location and must be labeled with dimensions (may be added in Fig. 1.
3. Results section is well-written, but the axis values, legend, and axis designations font size should be enlarged in all figures.
4. Section 4: More description of results, comparative, and critical analysis of the literature must be added to the discussion.
Author Response
Dear Editor and Referee,
Thank you very much for the appropriate comments and valuable suggestions on our manuscript. According to referee's opinions, we make a major revision on the manuscript, and would like to provide following response:
The responds to the reviewer’s comments are as following:
- Define Mie metamaterials in the introduction section?
Response: According to your suggestion, we added “In Mie resonance-based metamaterial, the electromagnetic interaction of dielectric particles with relative high permittivity can exhibit a strong magnetic or electric resonance based on Mie theory. From the oscillation of resulting magnetic or electric dipole, negative permeability or permittivity can be produced[15,16], and the frequency of magnetic or electric resonance shifts with shapes, sizes, dielectric constant and permeability of the dielectric cubes [17].” in our manuscript to difine Mie metamaterials. (Page 2/Line 54)
- Experimental section, Explain the addition or placement of metamaterial absorber material within the concrete. A schematic may be added to display the location and must be labeled with dimensions (may be added in Fig. 1.
Response: According to your suggestion, we added a schematic to display the location of metamaterial absorber material within the concrete as Fig.2(b).
- Results section is well-written, but the axis values, legend, and axis designations font size should be enlarged in all figures.
Response: According to your suggestion, we enlarged the axis values, legend, and axis designations font size in all figures.
- Section 4: More description of results, comparative, and critical analysis of the literature must be added to the discussion.
Response: According to your suggestion, more description of results and a table to compare the proposed structure with other types of absorbents on concrete were added in our manuscript. (Page 4/ Line 125, Page 5/ Line 180, Page 6/Line 190)
We have tried our best to improve the manuscript and made some changes in it. These changes will not influence its content and framework. We look forward to your further response. Thanks again for your comments!
With best regards,
Yours sincerely,
Hongya Wu
Round 2
Reviewer 1 Report
This reviewer thanks the authors for the improvement of the manuscript. At the present form, I recommend acceptance of the manuscript.